# Regulatory Potential of Competing Endogenous RNAs in Myotonic Dystrophies

**DOI:** 10.3390/ijms22116089

**Published:** 2021-06-04

**Authors:** Edyta Koscianska, Emilia Kozlowska, Agnieszka Fiszer

**Affiliations:** Institute of Bioorganic Chemistry, Polish Academy of Sciences, 61-704 Poznan, Poland; emiliak@ibch.poznan.pl (E.K.); agnieszka.fiszer@ibch.poznan.pl (A.F.)

**Keywords:** non-coding RNAs, microRNA, lncRNA, circRNA, ceRNA hypothesis, neuromuscular diseases, DM1, DM2, repeat expansion, disease modeling

## Abstract

Non-coding RNAs (ncRNAs) have been reported to be implicated in cell fate determination and various human diseases. All ncRNA molecules are emerging as key regulators of diverse cellular processes; however, little is known about the regulatory interaction among these various classes of RNAs. It has been proposed that the large-scale regulatory network across the whole transcriptome is mediated by competing endogenous RNA (ceRNA) activity attributed to both protein-coding and ncRNAs. ceRNAs are considered to be natural sponges of miRNAs that can influence the expression and availability of multiple miRNAs and, consequently, the global mRNA and protein levels. In this review, we summarize the current understanding of the role of ncRNAs in two neuromuscular diseases, myotonic dystrophy type 1 and 2 (DM1 and DM2), and the involvement of expanded CUG and CCUG repeat-containing transcripts in miRNA-mediated RNA crosstalk. More specifically, we discuss the possibility that long repeat tracts present in mutant transcripts can be potent miRNA sponges and may affect ceRNA crosstalk in these diseases. Moreover, we highlight practical information related to innovative disease modelling and studying RNA regulatory networks in cells. Extending knowledge of gene regulation by ncRNAs, and of complex regulatory ceRNA networks in DM1 and DM2, will help to address many questions pertinent to pathogenesis and treatment of these disorders; it may also help to better understand general rules of gene expression and to discover new rules of gene control.

## 1. Introduction

Non-coding RNAs (ncRNAs) form a group of distinctive RNA molecules that are not translated into proteins, but, instead, are responsible for important regulatory processes in cells [1,2]. Alterations in ncRNAs expression levels have been linked to a number of human diseases. There are several types of ncRNAs, characterized by specific biogenesis, function and activity, which basically differ in size; among them there are, for example, short microRNAs (miRNAs), long ncRNAs (lncRNAs) and circular RNAs (circRNAs). The small miRNAs contain a sequence that can be recognized by all types of RNA molecules. This sequence may be regarded as a "word" in the complex "RNA language" composed of specific letters—nucleotides. The ceRNA hypothesis assumes that all RNA molecules, both non-coding as well as protein-coding RNAs (collectively termed ceRNAs), communicate with and co-regulate each other by competing for binding to shared miRNAs [3]. There is growing evidence that, in pathology, the RNA communication system is disrupted [2,4,5,6,7,8]. It was also suggested that such communication between RNA molecules may play an important role in various repeat-associated diseases [9,10]. The common denominator of the repeat-associated diseases is a special mutation in relevant genes, where short nucleotide sequences are repeated many times and form unusually long tracts of repeats. It was proposed that repeat-associated ceRNAs (raceRNAs) characteristic of two neuromuscular diseases, namely, myotonic dystrophy type 1 and 2 (DM1 and DM2), may cooperate in their regulatory functions. Importantly, it has been shown that numerous miRNAs may bind to simple sequence repeats present in toxic RNAs and there are several reports addressing potential RNA cross-regulation in repeat-associated diseases [8,9,10,11,12,13,14]. Given that DM1 and DM2 are neuromuscular diseases, it seems additionally interesting to examine functional ncRNAs from the perspective of nervous and muscle cells, as the importance of ceRNA crosstalk in these tissues has been addressed [5,15]. All of this leads to the conclusion that in DMs a significant deregulation of the whole RNA regulatory network may occur, but at a level of complexity that requires further investigation.

## 2. A Brief Overview of ncRNAs and Their Integrated Networks

NcRNAs control every aspect of gene regulatory network activity, including transcriptional control, post-transcriptional processing and epigenetic targeting [16]. The best characterized ncRNAs are miRNAs, the main players in the ceRNA network. These endogenous small (21–23 nucleotides in length) RNAs control gene expression at the posttranscriptional level. The canonical biogenesis of animal miRNAs includes two subsequent RNA cleavage steps, namely, nuclear and cytoplasmic. Two RNase III endonucleases (Drosha and Dicer) process miRNA precursors sequentially to produce mature miRNAs. The miRNA biogenesis, the assembly of the miRNA-induced silencing complex (miRISC) and various intricate miRNA-mediated mechanisms of gene expression regulation have been discussed in detail [17,18]. The regulatory potential of miRNAs is enormous; to date, 2654 mature human miRNAs have been deposited in the miRNA repository (miRBase, Release 22) [19] and more than 60% of mammalian protein-coding genes may be regulated by miRNAs [20]. miRNAs downregulate gene expression by imperfect pairing with complementary sites within transcript sequences and repress gene expression by inhibiting protein synthesis that occurs with or without transcript degradation (reviewed in [21,22]). Specifically, miRNAs exert their regulatory effect mainly by binding to the 3’UTR of the target mRNAs, which results in translational suppression, mRNA deadenylation and decay, or mRNA cleavage [23,24]. The interaction between miRNAs and mRNAs is influenced by many factors; however, nucleotides 2–8 of the miRNA, termed the "seed" sequence, are essential for target recognition and binding [23].

The other group of regulatory RNAs is comprised of lncRNAs, that are molecules longer than 200 nucleotides. This is a very heterogeneous group of molecules; lncRNAs can be placed in one or more categories, depending on their genome localization and/or on their orientation (sense, antisense, bidirectional, intronic or intergenic lncRNAs) [16]. lncRNAs may contain open reading frames (ORFs) and are often transcribed by RNA polymerase II, spliced and polyadenylated. Originally, they were thought not to code for any protein product, but quite recent studies revealed that certain transcripts annotated as lncRNAs code for small peptides (micropeptides) with biologically important functions [25,26]. The genome-wide expression and evolutionary analyses suggest that some lncRNAs may play important functional roles; however, their cellular mechanisms of action are still largely unknown [27]. lncRNAs are transcribed in complex intergenic, overlapping, and antisense patterns relative to adjacent protein-coding genes, suggesting that many lncRNAs regulate the expression of these genes. Most lncRNAs are transcribed in a developmentally regulated and cell-type specific manner, particularly in the central nervous system (CNS), wherein over half of all lncRNAs are expressed [28]. lncRNAs have been shown to be implicated in many diseases, mostly cancers (reviewed in [29]), but also in neurodegenerative diseases, e.g., Huntington’s (HD) and Alzheimer’s (AD) (reviewed in [30]). Specific functions of lncRNAs have also been described in muscle cells [31]. Moreover, the mutual regulatory influence of mammalian miRNAs and lncRNAs has been reported [32]; some lncRNAs are degraded by miRNAs, others serve as sponges for miRNAs and a few lncRNAs were reported to compete with miRNAs for binding to mRNAs and to produce small RNAs. Direct links between the levels of miRNAs, mRNAs and lncRNAs have increasingly been discovered and abundant lncRNAs are currently thought to act as ceRNAs and miRNA sponges [33,34]. lncRNAs play key functional roles in controlling cellular regulation, most likely through their interactions with diverse classes of proteins; to date, however, the full spectrum of proteins that interact with ncRNAs is still unknown (reviewed in [35]).

The next ncRNA species with ceRNA activity are pseudogenes. Pseudogenes exist as either processed or non-processed genetic elements. They are almost as numerous as coding genes and represent a significant proportion of the transcriptome; however, very few pseudogenes have been functionally characterized so far [36]. Recent discoveries indicate that pseudogenes were retained during evolution because they possess various regulatory functions, which can be parental gene-dependent and -independent, may involve DNA, a sense or antisense mRNA, or may even be mediated by a protein [37]. Pseudogenes may play important roles in the ceRNA network; they are likely to act as perfect miRNA sponges, because they share many miRNA binding sites with their cognate genes [36,38].

The last but not least ncRNA group in the ceRNA interplay are circRNAs, a highly prevalent RNA species in the human transcriptome with emerging regulatory potential. Endogenous circRNAs are generated primarily through a type of alternative RNA splicing called “back-splicing”, in which a splice donor splices to an upstream acceptor, rather than a downstream acceptor. Some of circRNAs are abundant and stable in mammalian cells. They arise from both exons and introns, and they can act as scaffolds for proteins, recruit other RNA species and, through binding of miRNAs, can affect the transcriptional silencing, translation and/or decay of specific mRNAs (reviewed in [39,40]). In principle, two circRNAs with a strong potential to act as miRNA sponges have been reported, suggesting that circRNAs play important roles in regulating gene expression (reviewed in [41]); however, the global properties of circRNAs are not well understood. Moreover, it was demonstrated that circRNAs are highly abundant in the mammalian brain compared to other tissues, which highlights the role of circRNAs in the CNS [42]. Recently, it has been reported that circRNAs are involved in skeletal muscle myogenesis [43] and, most importantly, that global circRNA levels are increased in DM1 [44]. Overall, until now, thousands of circRNAs that are expressed in animal cells have been identified and characterized, providing new insights regarding their biogenesis, the cell-type specificity of their expression, the extent to which they are conserved, the extent to which they are translated and their potential to act as miRNA sponges [45].

## 3. Myotonic Dystrophies and Other Repeat-Associated Diseases

Microsatellites or simple sequence repeats (SSRs) are tandemly repeated 1–6 base pair (bp) long tracts of DNA that are ubiquitous in the genomes of all living organisms, in both protein-coding and non-coding regions (reviewed in [46]). Expansions of SSRs in specific regions of single genes can lead to various hereditary neurological diseases in humans. The trinucleotide repeat expansion diseases (TREDs) constitute the largest group of such disorders. This group includes disorders triggered by the expansion of a CAG repeat in the translated regions of respective genes, such as HD, six distinct types of spinocerebellar ataxia (SCA 1–3,6,7 and 17), dentatorubropallidoluysian atrophy (DRPLA) and spinobulbar muscular atrophy (SBMA). The TRED group also comprises fragile X syndrome (FXS) and fragile X-associated tremor/ataxia syndrome (FXTAS), both of which are caused by a CGG expansion in the 5’-untranslated region (5’UTR) of the *FMR1* gene, as well as Friedreich ataxia (FRDA) caused by a GAA expansion in the first intron of the frataxin (*FXN*) gene. Apart from the trinucleotide repeats, longer repeated motifs can also be associated with relevant diseases. Specifically, 5-nt repeat tracts of ATTCT and TGGAA are implicated in SCA type 10 and type 31, respectively. In addition, expansion of a hexanucleotide GGCCTG repeat causes SCA 36, while expansion of GGGGCC leads to the amyotrophic lateral sclerosis (ALS)/frontotemporal dementia (FTD) pathology. In view of this review, two repeat-associated neuromuscular diseases are of importance, namely, DM1 and DM2, because expanded repeat tracts in their mutant transcripts are most potent among all of pathogenic SSRs in influencing raceRNAs crosstalk [10]. In these disorders, the deleterious repeat expansions are located in non-coding regions of relevant genes. Specifically, the underlying molecular cause for DM1 is the existence of a pathological (CTG)n triplet expansion in the 3’UTR of the *dystrophia myotonica protein kinase (DMPK)* gene [47], whereas (CCTG)n repeats in the first intron of the *cellular nucleic acid binding protein/zinc finger protein 9 (CNBP/ZNF9)* gene cause DM2 [48]. As for the length of expanded repeat tracts, quite wide ranges were reported for DM1 and DM2 patients (Table 1). Classical DM1 is characterized by the range of 50–1000 CTG repeats; tracts longer than 800 may manifest as juvenile DM1 and tracts longer than 1000 as a congenital form of disease [49]. In the case of DM2, the mean CCTG repeat expansion size is around 5000 [50]. These repeat expansions are transcribed into (CUG)n- and (CCUG)n-containing RNA, respectively, which form stable secondary structures and sequester RNA-binding proteins, such as the splicing factor muscleblind-like protein (MBNL), forming nuclear aggregates known as foci (reviewed in [51]). A molecular mechanism of toxic RNA gain-of-function is characteristic of both DMs. Although the RNA toxicity and widespread spliceopathy are thought to be the major factors underlying the pathogenesis of DMs, alternative mechanisms, such as host gene haploinsuficiency, bidirectional transcription, additional changes in gene expression, translation efficiency, misregulated alternative polyadenylation, miRNA deregulation and non-canonical repeat-associated non-AUG (RAN) translation, may also contribute to the pathogenesis of these diseases [52,53,54,55,56,57,58]. Clinically, DM1 and DM2 are multisystem disorders characterized by myotonia, progressive muscle weakness, heart conduction defects, cognitive impairments, endocrine abnormalities, insulin resistance, cataracts, and gastrointestinal manifestations, with the symptoms being usually more severe in DM1 than in DM2. All the hallmarks of DMs, their clinical manifestations, and phenotypes, as well as various therapeutic perspectives, have been extensively reviewed [59,60,61,62,63,64,65]. Worthy of note is also an overview of miRNAs in the context of DM, as important roles of many miRNA have been linked with these diseases (comprehensively reviewed in [62]). Selected main features of DM1 and DM2 are presented in Table 1.

## 4. Transcripts Containing SSRs May Act as Potential ceRNAs

A hypothesis suggesting that transcripts bearing long sequence repeats might function as ceRNAs can be supported by different studies reported by us and others. Specifically, several transcripts that contain simple repeats of different nucleotide content and length have been reported to be under miRNA control (references, e.g., [11,82,83,84,85,86,87], and reviews, [88,89,90]). Importantly, we have thoroughly addressed the issue of regulation of the *DMPK* gene expression by miRNAs. We showed that miRNAs may interact both with typical miRNA binding sites and expanded CUG repeats present in the *DMPK* 3’ UTR (the latter interaction will be discussed in detail in the next chapter). Moreover, we reported that miRNAs may cooperate in *DMPK* silencing; both typical miRNAs located at a cooperativity-permitting distance and the CUG-repeat-binding miRNAs may act cooperatively. In another study, we have demonstrated that mutant *HTT* expression can be selectively inhibited by targeting the expanded CAG repeats with exogenously delivered short interfering RNAs (siRNAs) functioning as miRNAs [91]. The use of miRNA-like siRNAs resulted in a very efficient selective silencing of the mutant allele, due to cooperative action of adjacent miRNA-induced silencing complexes (miRISCs) on the expanded CAG tract [92]. Most importantly and most recently, we have comprehensively studied a potential role of extended SSRs in ceRNA crosstalk [10]. We applied bioinformatics tools and we identified human transcripts that may be regarded as raceRNAs. We also identified multiple protein-coding transcripts, transcribed pseudogenes, lncRNAs and circRNAs showing this potential and predicted numerous miRNAs that may bind to SSRs. We proposed that simple repeats expanded in various hereditary neurological diseases may act as sponges for miRNAs containing complementary repeats that would affect raceRNA crosstalk. Importantly, miRNA sequestration on expanded microsatellite RNA has been also suggested for FXTAS, another repeat-associated disease. Specifically, the presence of identical miRNA recognition elements (MREs), on both misregulated mRNAs and the expanded microsatellite RNA, was reported [12], which corroborates that miRNA sequestration may lead to miRNA depletion and upregulation of the respective target mRNAs.

## 5. Current Understanding of the Activity of ceRNAs in DMs and Directions for Further Research

An understanding of the role and importance of ncRNAs in numerous human diseases, including DM1 and DM2, is increasingly growing. Several types of ncRNAs, such as miRNA, lncRNA or circRNA, are affected in both DMs, suggesting that they are implicated in multiple disease mechanisms at the molecular level [44,62,66,93,94,95,96,97] (Table 2). Moreover, changes in ncRNA expression profiles were reported mostly in muscle cells, the primary cell type affected in DMs, which additionally implies their involvement in the pathophysiology of the disease. However, possible contributions of ceRNA functions have not been fully described.

The RNA gain-of-function has a primary role in the DM pathogenesis. One explanation of how repeat-containing RNA can cause disease symptoms is through interaction with RNA-binding proteins, MBNL1 and CUGBP Elav-like family member 1 (CELF1), which regulate multiple RNA-processing events, including alternative splicing [53,56,98]. Both MBNL1 and CELF1 were shown to be under miRNA control [79,99,100] and they are also important proteins that may target or interfere with specific ncRNA regulatory pathways. Importantly, MBNL regulates miRNA biogenesis [78] and circRNA formation [101]. Moreover, extensive crosstalk between miRNAs and splicing factors in regulating alternative splicing and gene expression program, during development and cellular differentiation, was reported [102]. Because individual miRNAs can target multiple miRNAs [23] and different RNA regulatory processes are interrelated and overlapping, we can expect that ceRNA networking and miRNA sponging may play an important role in the pathogenesis of both DMs. Of great importance are the facts that myocyte enhancer factor 2C (MEF2C), which is a transcription factor regulating heart and skeletal muscle differentiation and growth, was shown to regulate the quantity and quality of the microtranscriptome [103] and the MEF2 transcription network was shown to be disrupted in DM heart tissue, which dramatically alters expression of a large number of miRNAs and mRNA targets [79].

The vast majority of studies supporting involvement of ceRNA activities in DMs focus on miRNAs as key factors contributing to disease pathogenesis. Global changes in miRNA expression patterns in DM1 and DM2 have been reported, as well as deregulation of muscle-specific miRNAs (myomiRs) [73,81]. The myomiRs (miR-1, miR-206 and miR-133a/b) are highly and specifically expressed during cardiac and skeletal muscle cell differentiation, with miR-206 being the only myomiR specific to skeletal muscle (reviewed in [96]). The role of miRNAs in muscle development has also been shown in many studies [106,107,108]. Importantly, the deregulation of DM1/DM2-associated miRNAs has been linked to alterations in their putative target expression, indicating pathological potential of miRNA dysregulation. A current picture of miRNA dysregulation in DM has been presented in [62], where different studies addressing the role of miRNAs in both DM1 and DM2 were discussed. Moreover, it was emphasized that pathological role of altered miRNAs displayed by DM patients makes them good biomarkers and also novel therapeutic targets [62].

The importance of other ncRNAs, with a particular focus on lncRNAs and circRNAs, in the context of DM pathology has also been highlighted in the literature [44,93,96,105,109] (Table 2). In the last years, lncRNAs are emerging as critical regulators of muscle differentiation, growth, and regeneration, as well as important factors contributing to muscle disease [96,109,110,111,112,113]. Relevant functions of lncRNAs in muscle cells were reported, e.g., linc-MD1 acting as a sponge for specific miRNAs important for regulation of transcription factors that activate muscle-specific gene expression [31]. Moreover, it was shown that lncIRS1 acts as a sponge for the miR-15 family to regulate insulin receptor substrate 1 (IRS1) expression, resulting in promoting skeletal muscle myogenesis and controlling atrophy [114].

As for circRNAs, numerous studies have also verified their gene regulatory functions in skeletal muscle development, including ceRNA mechanisms [39,43,115,116,117]. Together, recent findings on functions of lncRNAs and circRNAs in skeletal and cardiac muscles biology were extensively reviewed in [15,109], where their nuclear and cytoplasmic activities are also discussed, as well as potential crosstalk between miRNAs, circRNAs and lncRNAs.

Generally, deregulation of circRNAs has been associated with a muscle pathological state. It was shown that several circRNAs were deregulated during myoblast proliferation and muscle cell development [118]. Nevertheless, of particular interest are links between aberrations in circRNA levels and DM1 pathogenesis. Importantly, in one investigation, circRNAs expressed in DM1 skeletal muscles were identified by analyzing RNA-sequencing data-sets followed by qPCR validation [105]. In muscle biopsies, four circRNAs, out of nine tested, were upregulated, compared to healthy controls (circCDYL, circHIPK3, circRTN4_03 and circZNF609). Little is known of the identified DM1-circRNAs, but this study provides the first evidence that the levels of specific circRNAs linked to myogenesis are deregulated in skeletal muscle biopsies and in myogenic cell cultures derived from DM1 patients. In another study, a global increase in circRNA levels in DM1 was reported and numerous circRNAs increases in DM1 were identified [44]. It was a very important but striking observation, because, rather, deregulation of some circRNAs was expected, due to diminished functional levels of MBNLs sequestered in mutant RNA foci. However, no deregulation of the analyzed circRNAs in muscle was observed in DM1 and DM2 samples, when compared to non-DM samples. Instead, a subset of circRNAs that were upregulated in DM1 samples was identified. Moreover, these elevated circRNA levels were associated with muscle weakness and alternative splicing changes, that are biomarkers of DM1 severity. The role of the increased level of circRNAs in the pathogenesis of DM1 is unknown and requires further investigation. Interestingly, one of the elevated circRNA was circZfp609, which is supposed to play a role in promoting myoblast proliferation, possibly by sponging miR-194-5p [115,119].

This review is meant to bring new background knowledge for RNA cross-regulation in DMs. Based on our previous and ongoing research [10,11], as well as on published conclusions [2,33,120], we propose that the mutant *DMPK* transcripts serve as molecular sponges for natural miRNAs having CAG repeats in their seed sequence, sequestering them and thereby preventing them from their physiological activity. We showed that miRNAs with CAG repeats in their seed regions can bind to the CUG repeats present in the *DMPK* transcript [11]. More specifically, we showed that some of the CUG-repeat-binding miRNAs may act cooperatively to down-regulate *DMPK* expression (miR-15b/16, miR-214) and the degree of miRNA-mediated repression increases with the length of the repeated sequence in the *DMPK* 3’UTR. Moreover, we have found miR-16 in cytoplasmic foci formed by exogenously expressed RNAs with expanded CUG repeats. Therefore, we hypothesize that the sequestration of CUG-repeat-binding miRNAs makes them inactive, or simply prevents them from regulating other transcripts containing these miRNA binding sites, leading to a more widespread impairment of the miRNA-mediated regulation of gene expression. The work by Witkos et al. on RNA crosstalk in repeat-associated diseases is the first report addressing potential ceRNA activity in DM1 and DM2 [10]. Extensive bioinformatics analyses of putative ceRNAs containing short tandem CUG and CCUG repeats and their interactions with miRNAs bearing sequence repeats revealed that the raceRNA crosstalk could be perturbed in DM1 and DM2; however, these potential perturbations have not been experimentally examined yet. Interestingly, approximately 120 circRNAs containing at least 5 consecutive DM1-relevant repeats were identified. When miRNA binding to various SSR tracts was examined, the greatest number of miRNAs was predicted to bind to CUG and CCUG tracts; 19 and 25 miRNAs showing complementarity within their seed regions were predicted to bind to DM1- and DM2-relevant SSRs. Several of these CUG- and CCUG-binding miRNAs were highly expressed in muscle. Moreover, abnormally elongated mutant transcripts with the expansion of CUG and CCUG tracts in patients with DM1 and DM2, respectively, appeared to influence miRNA crosstalk in myoblasts, as repeats found in these patients were predicted to have a great impact on MRE site occupancy. Predicted effects of the CUG and CCUG tract expansions on raceRNA crosstalk in DM1 and DM2 should cause deregulation of the expression of genes that exhibit MREs for miRNAs interacting with these repeats. Consequently, this would lead to elevated levels of transcripts containing these MREs. Overall, expression of elongated CUG and CCUG tracts can lead to more global de-repression of miRNA-mediated gene regulation and a physiological ceRNA network in DM1 and DM2 can be altered by pathological repeat expansions.

Different pathogenic mechanisms exist to explain how repeat expansions in the genome of affected patients lead to the DM phenotype [53,56,58,98]. Despite strong evidence that RNA-binding proteins play a pivotal role in DM1 and DM2 pathologies, the downstream pathways controlled by these RNA-binding proteins are not fully understood. In this regard, a further study of ceRNA crosstalk in these diseases is of special importance, as other RNA-based regulatory mechanisms may contribute to the development of the disease. The expanded CUG and CCUG repeat tracts present in mutant transcripts typical of DMs may be potent miRNA sponges for miRNAs with specific repeats in their seed regions and may affect their physiological functions. Thus, a possibility of miRNA sponging seems to be a significant factor, which contributes not only to disease pathogenesis, but also to global changes at RNA levels and deregulation of total gene expression. The general concept of miRNA-mediated crosstalk between transcripts containing expanded CUG and CCUG repeats in relation to DM1 and DM2 is presented in Figure 1.

Regarding putative perturbations in raceRNA crosstalk and relevance of miRNA sponging in the pathogenesis of DM, further studies are needed to explain whether the presence of expanded repeats and aberrant expression of certain ceRNAs are interrelated. Interestingly, it was recently reported that functional mechanisms of lncRNAs and the potential pathogenic mechanisms of expanded microsatellite RNA may be linked [9]. Shared mechanisms include protein sequestration, peptide translation, miRNA processing and miRNA sequestration. However, it should be also noted that different ceRNAs may bind and sequester miRNAs with unequal efficiency, resulting in different miRNA-mediated target repression. Moreover, the level and accessibility of endogenous ceRNAs and the endogenous mRNA targets are very important for the functional outcome. Factors affecting ceRNA activity, as well as the conflicting conclusions of recent ceRNA studies, were discussed in [43].

For potential activity of mutant *DMPK* and *CNBP* transcripts in the ceRNAs network, a mechanism of somatic expansions occurring typically in non-mitotic cells of DM patients is also important [121]. An incorrect DNA repair, coupled to transcription across the repeat tract, may result in a much higher number of repeated units in skeletal muscle and brain cells than observed initially in the mutant gene [122]. Abnormal abundance of MREs for repeat-binding miRNAs may additionally enhance the miRNA sponging potential of raceRNAs.

Considering the role of miRNA sponging in DMs, one more important aspect should be taken into account, i.e., potential intracellular co-localization of miRNAs and transcripts with repeat tracts. Although the prevailing view is that miRNAs execute their function in the cytoplasm and the expanded CUG and CCUG repeats form nuclear foci, there are reasons to believe that important repeat-binding miRNAs could be sequestered by the repeats. First, during cell division, some miRNAs could be sequestered by the repeats. Second, it was shown that miRNA, together with Argonaute 2, can localize in the nucleus, suggesting that nuclear miRNA may also regulate protein expression at the level of DNA. Moreover, some miRNAs are present in equal concentrations in both nuclear and cytoplasmic compartments, are somewhat nuclear enriched, or can be differentially expressed under stress conditions [123,124,125,126,127]. Importantly, nuclear enriched are for example miR-15b and miR-16, which are top CUG-repeat-binding miRNA candidates, chosen based on the number of matches with the CUG repeats in their seeds, as well as on experimental validation [11].

Interestingly, various types of ncRNAs are found in exosomes, which are small extracellular vesicles (EVs) that serve as mediators of cell-to-cell and tissue-to-tissue communication, and, therefore, may provide an additional level of regulatory ceRNA crosstalk. Exosomal ncRNAs are detected in a variety of body fluids, but so far, they have been investigated mostly in cancer [128]. However, it has been shown that exosomes carrying four muscle-specific miRNAs (myomiR-1, -133a, -133b, -206) are elevated in the blood of muscle disease patients and, most importantly, that the level of circulating myomiRs in blood of DM1 patients can be associated with the progression of muscle wasting [72,129].

Finally, given that DM1 and DM2 are neuromuscular disorders, it seems additionally interesting to consider ceRNA networks also in the context of other neurological diseases. The molecular basis of DMs in the nervous system has just started to be revealed [130,131]; MBNL1 and probably CELF1 may both be involved in CNS alterations, but little is known about molecular defects causing highly variable CNS symptoms in DMs [132,133]. It cannot be ruled out that various ceRNA mechanisms in the brain may be also implicated in the pathogenesis of these diseases. Importantly, ncRNAs are particularly abundant in CNS and their expression is dynamically regulated [134]. One interesting example is complex regulation of *BACE1* expression, in which specific antisense transcripts, lncRNAs and miRNAs, are operating in AD [135,136]. Among diseases caused by repeat expansions, SCA 7 is the example where specific ceRNA network was identified and its disruption described as crucial for retinal and cerebellar neurodegeneration [13]. This network includes miR-124 interaction with lnc-SCA7, which affects expression level of the mutant *ATXN7* gene responsible for SCA 7.

Despite many unknowns regarding ceRNAs functioning and some controversy surrounding the general concept of ceRNA regulation under physiological conditions, this type of RNA cross-regulation in DMs is worth investigating in more depth. Most importantly, there is promising evidence that potential miRNA sequestration and changes in miRNA expression patterns could be a reliable diagnostic tool. Moreover, as mentioned earlier, miRNA may serve as potential biomarkers helping in diagnosis and/or prognosis of the disease, as well as novel therapeutic targets [8,62,72,137,138,139]. In one of the recent studies miR-7, downregulated in DM1, was described as crucial for muscle functioning; restoration of miR-7 levels rescued pathogenic effects observed in DM1 myoblasts, making this miRNA a candidate therapeutic target [140]. The possibility of utilizing lncRNAs as therapeutic targets for skeletal muscle disorders in humans is also considered [113]. Moreover, a possible future use of circRNAs as biomarkers of DM1 severity was reported [44,105]. Generally, different elements of a disrupted ceRNAs network could be potentially targeted in various therapeutic approaches. Effective therapeutic strategies for targeting ncRNAs have been developed and mainly chemically modified oligonucleotides are used for this purpose [141]. Overall, promising evidence exists that more ncRNA-based diagnostic and therapeutic applications will emerge in the future. However, to use them successfully in the context of DMs, it is necessary to precisely identify the altered ceRNAs interactions and their potential contribution to neuromuscular dysfunction.

## 6. Advanced In Vitro Models for Studying the ceRNA Crosstalk in DMs

Extending knowledge of gene regulation by ncRNAs, and of complex regulatory ceRNA networks in the case of DM1 and DM2, will help to address many questions related to the pathogenesis and treatment of these disorders. However, the effects of ceRNA activity need to be modelled in the real scenario in cells, where many miRNAs interact with many targets, not only a single miRNA interacting with some targets.

In line with animal models, cultured cells showed to be an essential model for both fundamental and translational research on DM [142]. Different in vitro cell models were developed and successfully used to study disease-related molecular mechanisms and evaluate therapeutic approaches before in vivo validation. In this review, we focus on the usefulness of advanced in vitro DM models in studying pathological potential of RNA dysregulation, as well as specific interactions between various types of ceRNAs. However, the use of appropriate cellular models of DM1 and DM2 for different ceRNA crosstalk investigations is considerably limited due to the neuromuscular nature of these diseases, low levels of DMPK and CNBP/ZNF9 transcripts in affected cells [97,143,144] and genetic background variation [145]. This limitation can be overcome by the application of genome editing tools and induced pluripotent stem cell (iPSC) technology. The resulting iPSCs are similar to embryonic stem cells in terms of self-renewal and the potential to differentiate into any cell type including neurons and muscles cells. The iPSCs can be easily obtained from somatic cells (e.g., fibroblasts) in the process called reprogramming, by delivery of just four reprograming factors, which enables to restore an embryonic state of cells. Moreover, sequestration of MBNL family splicing factors, which is observed in DMs, facilitate reprograming, as these protein factors are identified as negative regulators of a pluripotency state [146,147]. All the mentioned properties make the iPSCs an invaluable tool for modeling diseases, but also for drug testing or cell-based therapies, especially in light of the use of rapidly developing genome editing technologies [148,149].

Difficulties in obtaining viable cardiac and nervous tissues make it hard to model DMs at a cellular level. Moreover, primary cultures do not mimic a developmental stage of these diseases. Alternatively, iPSC technology is a renewable source of cells that can help to understand whether the disease arises from developmental or degenerative processes. Cells from one patient can be hypothetically differentiated into any type of cells (excluding reproductive cells), which means that it is possible to obtain nerve cells, muscle cells and other types of cells with the same genetic background. However, it should be taken into account that the maturation of human iPSC-derived skeletal muscle fibers using current in vitro protocols is generally limited and needs to be improved.

Currently, in the DM1 and DM2 field, most of researchers focus on differentiated cells, such as neurons or muscle cells, commonly affected in these diseases. However, in contrast to other repeat expansion diseases, some DM hallmarks, such as nuclear RNA foci, are observed even in not specialized immature iPSCs [150]. Aberrant splicing patterns, being a consequence of presence of nuclear foci containing mutant DM1 mRNA and sequestered splicing factors, such as MBNL family proteins, were found in iPSC-derived neurons and astrocytes [151] and also in iPSC-derived myogenic cells, such as myocytes and myotubes [150] and cardiomyocytes [152]. The cardiomyocytes of DM patients exhibited some calcium-handling defects, potentially resulting from cardiac complications reported in DMs. In case of DM2, only RNA foci were observed, but no aberrant splicing patterns connected with sequestration of MBNL proteins. Moreover, RNA sequencing (RNA-seq) of DM1 and DM2 iPSC-derived cardiomyocytes identified unique aberrant splicing and gene expression profiles in DM1 and DM2 cells, which suggests other mechanisms of pathogenesis [152]. Importantly, various pathological phenotypes observed in DMs can be reversed by correction of CUG repeats’ lengths in the iPSCs and iPSC-derived cells by genome editing techniques [150,153,154,155]. This possibility is of great significance in practical applications. Corrected cells have potential to be used not only in different cell-based therapies, but also in all approaches that require simultaneous formation of isogenic control cell lines, i.e., cells with the same genetic and epigenetic backgrounds as the cells with a specific mutation [150,153]. Alternative genome-editing strategies with therapeutic potential involved introduction of a premature poly(A) addition site upstream of CTG repeat tract of *DMPK* gene in the iPSCs [154,155] and the iPSC-derived neural stem cells (NSCs) [153,155]. This modification resulted in reduction of nuclear RNA foci and returning aberrant alternative splicing pattern to the physiological state, both in differentiated and non-differentiated cells [153,154,155]; however, epigenetic abnormalities across the mutant DM1 locus cannot be directly corrected by this strategy.

There is yet another aspect to be aware of, since repeat instability is frequently observed in patient-derived cells. Specifically, CTG repeats are highly unstable in cells both during reprogramming processes and subsequent passages of iPSCs [156,157]. In the case of DM1 iPSCs, a length limit of 126 CTG repeats appears to be important for reprogrammed cells and, generally, the expansion rate increases dramatically with the repeat tract’s length [156]. The differentiation process of iPSCs is supposed to reduce instability of repeats [156]; however, some authors suggest that the observed effect is a result of slower divisions of non-iPS cells, rather than inhibition of a repeat expansion process [157].

Overall, the combination of two powerful technologies, namely, genome editing and iPSC technique, have opened a new era for human-relevant disease modeling, drug testing and regenerative medicine. However, there are still some limitations to be addressed when using these methods. For example, difficulties in recapitulating the complexity of the entire organism or organs could be partially overcome by the development of organoid-based methods [158]. Recently, it was also shown that it is possible to obtain an advanced 3D model of DM1 human skeletal muscle that can be used in a preclinical platform for DM1 drug development [159]. Nevertheless, implementation of the above-mentioned modern technologies and solutions is of paramount importance to create adequate cellular models of DM1 and DM2 that will be useful to study complex RNA-RNA interactions. Identification and validation of different interactions between ceRNAs in models that do not fully reflect pathomechanisms of DMs may lead to misinterpretation of findings or incorrect results, especially in the context of low DMPK and CNBP/ZNF9 transcript levels in affected DM patients’ cells. Of special interest is to use human iPSC lines and the CRISPR/Cas system to create isogenic lines and, therefore, eliminate background genetic variation that could affect the expected outcome [145]. The appropriately engineered isogenic DM cell lines that provide genetically matched control cells could be a reliable model to study various ceRNA interactions, for example, by switching on and off individual components of the ceRNA network, potentially operating in DMs.

## 7. Conclusions

The substantial evidence supporting the ceRNA hypothesis is still missing; to date, there is only one study reporting functional circRNA and a physiologically relevant ceRNA mechanism in mammals [160]. There is also a degree of controversy over some of the biological relevance of this hypothesis [161,162,163,164]; however, it was proposed that effective competitor RNAs should be highly abundant in the cell or should contain multiple binding sites for a single miRNA species. In this scenario, the long CUG and CCUG repeats extended in DM1 and DM2, respectively, appear to be very strong candidates. Nevertheless, the involvement of expanded C/CUG repeat-containing transcripts in miRNA-mediated RNA crosstalk in cells has not been addressed experimentally.

It was suggested that ceRNAs can be connected to each other by direct or indirect connections; that is, two ceRNAs may share binding sites for a common miRNA (direct connection), or they can be connected through a third ceRNA (indirect connection) [165]. Therefore, for modelling the effects of ceRNAs in the post-transcriptional regulation of gene expression in cells, the effects of directly and indirectly connected ceRNAs need to be considered. Moreover, the effects of many miRNAs interacting with multiple targets should be examined. To study these complex RNA–RNA interactions in DMs, it is extremely important to use adequate methods [166] and models of DM1 and DM2 [142]. Of special interest is to use human iPSC lines and genome-editing technologies in order to create isogenic cellular lines and therefore eliminate background genetic variation that could affect the expected outcome. The use of isogenic cell lines is of paramount importance as genetic background differences between these individuals, even when controlled for by age, sex, and ethnicity, can negatively affect the results, when trying to identify specific mechanisms [145]. Moreover, therapeutic genome editing for DM1 and DM2 that enables to eliminate toxic RNA C/CUG repeats is extremely needed and potentially fruitful.

Given the complexity and degree of interactions between miRNAs and their various targets, being either protein-coding or non-coding transcripts, a better understanding of the rules governing gene regulation by ncRNAs and the intricate mechanisms that are involved in disease pathogenesis is necessary. Perturbations of ceRNAs and ceRNA networks could have consequences for diseases, but may also help to explain disease processes and present opportunities for new therapies.

## Figures and Tables

**Figure 1 ijms-22-06089-f001:**
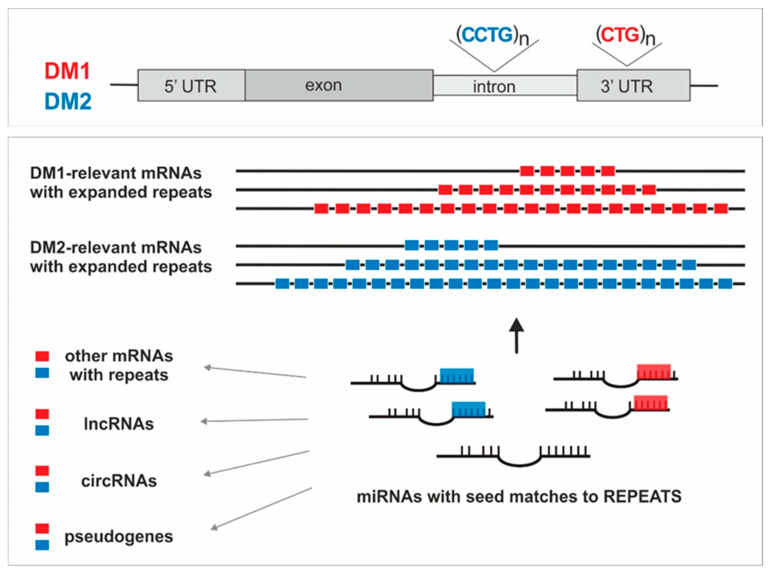
The concept of miRNA-mediated crosstalk between transcripts containing repeat tracts in relation to DM1 and DM2. Genes mutated in DM1 and DM2 with location of repeats are schematically presented (top panel). Potential interactions between different types of raceRNAs are depicted, as well as miRNAs interacting with DM1/DM2- relevant transcripts with expanded repeats of exemplary lengths (bottom panel).

**Table 1 ijms-22-06089-t001:** Main features of DM1 and DM2.

Features	DM1	DM2
Chromosomal locus	19q 13.3	3q 21.3
Gene expansion	DMPK (CTG)_n_	CNBP/ZNF9 (CCTG)_n_
Normal repeat size	Up to 37	Up to 27
Expanded repeat range	50–4000	75–11,000
Age of onset	At any age	At adulthood
Clinical manifestation	Refs. [59,60,66,67]	Refs. [59,60,66,67,68]
Altered miRNA	Refs. [69,70,71,72,73,74,75,76,77,78,79,80]	Refs. [78,81]

**Table 2 ijms-22-06089-t002:** Examples of deregulated ncRNAs in DMs.

ncRNA	Deregulation Reported in DM1/DM2	Reference
miRNA	DM1: miR-206, miR-1, miR-335, miR-29b, miR-29c, miR-33, miR-33a, miR-23a/b, miR-191, miR-208a, miR-7, miR-10, miR-133a/b, miR-15a, miR-22, miR-155	[62]
	DM2: miR-221-3p, miR-34c-5p, miR-208a, miR-381, miR-34b-3p, miR-34a-5p, miR-146b-5p, miR-193a-3p, miR-193b-3p, miR-125b-5p, miR-378a-3p, miR-1	[62]
lncRNA	MALAT1, DM1-AS	[94,104]
circRNA	circCDYL, circHIPK3, circRTN4_03, circZNF609, circGSE1, circFGFR1, circCAMSAP1, circBNC2, circZfp609, circHipk3	[44,105]

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
