# Peer review of "Regulatory Potential of Competing Endogenous RNAs in Myotonic Dystrophies"

_ijms, 2021, doi:10.3390/ijms22116089_

Round 1
Reviewer 1 Report
The potential involvement of ncRNA in the pathogenesis of DM1 and DM2 is of wide interest. The authors of this review describe the involvement of ceRNA in DM1 and DM2 but the specific contributions of ceRNA in DM and potential links to pathogenesis are underdeveloped. The reader may not be familiar with the ceRNA hypothesis in DM and thus a more detailed description of how ceRNA could directly impact pathogenesis is needed. The authors should address the following points.
- A clear description of how ceRNA is proposed to drive or contribute to pathogenesis in DM is required with more specific examples from the literature. With the exception of a number of their studies, there is not sufficient specific details describing other work from the field in support of the ceRNA hypothesis.
- The figure does not add much to the review and should provide more details, perhaps with specific examples of deregulated ncRNA reported in DM1 and DM2 from the different classes.
- Lines 32-34: “There are several types of ncRNAs that basically differ in size;”. This is a very broad generalization that should be reworded since the biogenesis, function and activity of these broad classes of ncRNA are distinct.
- Lines 34-35: “The small miRNAs are particularly important…” and Lines 77-78 “The other important group of regulatory RNAs…” The use of the word important in this context should be avoided as there is not sufficient understanding of each of the rolls of these RNA to compare importance.
- Lines 81-83: “LncRNAs may contain open reading frames…” It would be good to comment on the evidence of functional micropeptides coded by what were thought to be lncRNA (Nelson et al. 2016 Science, Chen et al, 2020 Science).
- Line 130: “Molecular Dystrophies” should be “Myotonic Dystrophies”
- Lines 288-290: “Different pathogenic mechanisms…”. There is a clear link between MBNL sequestration, mis-splicing of its targets and specific phenotypes observed in DM patients such as myotonia, cardiac conduction defects and insulin insensitivity. While other mechanisms are not clearly linked to pathogenesis, the role of MBNL sequestration is well documented in the literature. The authors should address hos miRNA cross-talk and ceRNA influence the transcriptome deregulation resulting from MBNL sequestration which may modify pathogenesis or alternatively contribute independently to pathogenesis.
- Line 380: “Unlimited” should be edited to “renewable”.
The section: “Advanced In vitro Models for Studying…” should focus more directly on how these systems could benefit the study of ceRNA or ncRNA function specifically in DM rather than describing the general usefulness of the tools.
Reviewer 2 Report
This review summarizes the current knowledge on the involvement of ncRNAs in the pathomechanism of myotonic dystrophies. It is comprehensive but could use more figures and tables to make it easier to understand.
The following is a list of what I believe to be typos.
L130. “Molecular dystrophies” is a typo for “Myotonic dystrophies.”
L206. “circrcRNA” is a typo for “circRNA.”
Reviewer 3 Report
The manuscript entitled "Regulatory Potential of Competing Endogenous RNAs in Myotonic Dystrophies" discusses the role of ncRNAs in myotonic dystrophies and the trinucleotide repeat expansions involvement in competing endogenous RNA crosstalk. The work merits potential interest to readers, but the work lacks information on specific works describing ncRNAs altered in myotonic dystrophy. Many things need to improve for the publication of high-quality work in IJMC. In my opinion, this article could be publishable after addressing these major and minor concerns.
#Major 1. some original studies on altered miRNAs in disease should be included. For example, in Table 1, several reviews are cited instead of the original works describing alterations in miRNAs. These works should be included in the table as well as discussed in the main text.
Rau F, et al. 2011 DOI: 10.1038/nsmb.2067
Fernandez-Costa J M, et al. 2013 DOI: 10.1093/hmg/dds478.
Kalsotra A, et al. 2014 Jan 30;6(2):336-45. DOI: 10.1016/j.celrep.2013.12.025
Fritegotto C, et al. 2017 DOI: 10.1007/s10072-017-2811-2
Koutsoulidou A, et al. 2015 DOI: 10.1371/journal.pone.0125341
Cappella M, et al. 2018 DOI: 10.1038/s41419-018-0769-5
#Major 2. The authors should add a new table summarizing all the ncRNAs altered in the disease reported to date and the original works' references. This table would be of great use to readers and would add value to the work.
#Major 3. The potential of ncRNAs as biomarkers of disease is discussed in the manuscript. However, it is not discussed that these ncRNAs may be new therapeutic targets for the disease. For example, recent work shows that the recovery of the levels of a specific altered miRNA in disease was sufficient to rescue DM1 myoblast defects (Sabater-Arcis M 2020. doi: 10.1016/j.omtn.2019.11.012). It would also be very compelling to discuss the possibilities of blocking the crosstalk of ceRNAs as a therapeutic strategy.
Minor #1. Circulating ncRNAs are usually found in exosomes. Authors should consider including exosome studies in the manuscript.
Minor #2. In the section "Advanced In Vitro Models for Studying the ceRNA Crosstalk in DMs", the author discusses the potential of iPSC technology in a very successful way. They also point to the need of the development of organoids to recapitulate the complexity of organs. In this direction, it has been very recently published a new advanced model for DM1 consisting of bioengineered 3D muscle structures (Fernandez-Garibay X, et al. 2021 DOI: 10.1088/1758-5090/abf6ae). I suggest that the authors review this work in this section of the manuscript.
Minor #3. Recheck all representations in the manuscript for typos, for instance, "cicrcRNA" in line 206.
Round 2
Reviewer 1 Report
The authors have done a commendable job in addressing concerns and the addition of the new table provides useful information.
Reviewer 3 Report
I would like to thanks the authors for this revised version of the manuscript. The authors made substantial changes following the reviewer's suggestions. Thus, I recommend this article be publishable.